# Targeted distribution of long-lasting insecticidal nets by community health workers to sustain household coverage: A pilot feasibility study in Western Uganda

**Annika K. Gunderson**[1,2], **Rapheal Mbusa**[3], **Emmanuel Baguma**[3], **Emmanuel Ayebare**[3], **Dana Giandomenico**[1], **Raquel Reyes**[4], **Moses Ntaro**[3], **Edgar M. Mulogo**[3], **Ross M. Boyce**[1,2,5]*

**1** Department of Epidemiology, Gillings School of Global Public Health, University of North Carolina at Chapel Hill, Chapel Hill, North Carolina, United States of America, **2** Carolina Population Center, University of North Carolina at Chapel Hill, Chapel Hill, North Carolina, United States of America, **3** Department of Community Health, Faculty of Medicine, Mbarara University of Science & Technology, Mbarara, Uganda, **4** Division of Hospital Medicine, University of North Carolina at Chapel Hill, Chapel Hill, North Carolina, United States of America, **5** Institute for Global Health and Infectious Diseases, University of North Carolina at Chapel Hill, Chapel Hill, North Carolina, United States of America

* ross_boyce@med.unc.edu

## Abstract

Universal coverage is defined by the World Health Organization as 1 long-lasting insecticidal net (LLIN) for 2 people in a household. While Uganda has been a leader in the distribution of LLINs, there are concerns regarding the longevity of LLINs. The main aim of this study was to address the LLIN coverage gap that emerges in the period after mass distribution campaigns through the implementation of a novel LLIN distribution strategy utilizing the existing community healthcare worker (CHW) infrastructure. We conducted a pilot feasibility study in two villages randomized to be the control or intervention. CHWs in both villages carried out their regular duties and calculated household eligibility to receive LLINs, classified as having a child under five positive for malaria and being below universal coverage. Only CHWs in the intervention village distributed LLINs to eligible households to reach universal coverage. Summary statistics were calculated for intervention implementation and malaria outcomes. Structured interviews were conducted with CHWs to assess burden and community acceptability of the intervention. Of the children evaluated by the CHWs, 102 of 169 (60.3%) and 112 of 171 (65.5%) were tested for malaria, of which 62 (60.7%) and 71 (63.3%) tested positive in the intervention and control villages, respectively. Only three households were at universal coverage. There was an increase from 4.0% to 6.5% of households meeting universal coverage in the intervention village, compared to a decrease from 7.8% to 1.8% in the control village after the follow-up period. There was an increase in the number of children under the age of 5 who slept under an LLIN the previous night from 15.7% to 31.6% in the intervention village compared to a decrease in the control village from 29.1% to 10.5%. No CHWs reported an increased burden from the intervention and all reported favorable opinions. Our pilot study demonstrates the feasibility and acceptability of targeted LLIN distributions leveraging the

**Data availability statement:** Deidentified individual data that supports the results will be shared following publication provided the investigator who proposes to use the data has approval from an Institutional Review Board (IRB), Independent Ethics Committee (IEC), or Research Ethics Board (REB), as applicable, and executes a data use/sharing agreement with UNC. The Principal Investigator should submit a request to the UNC Industry Contracting team (OSPContracting@unc.edu) to initiate the data use/ sharing agreement.

**Funding:** The work was funded by an award from the Conservation, Food, and Health Foundation (RMB). Support was also provided from the National Institute of Allergy and Infectious Diseases to RMB (K23AI141764) and AKG (T32AI070114). Additionally, data collection for the project was supported by the National Center for Advancing Translational Sciences (NCATS), National Institutes of Health (UM1TR004406). The funders had no role in study design, data collection and analysis, decision to publish, or preparation of the manuscript.

**Competing interests:** The authors have declared that no competing interests exist.

existing structure to supplement national distribution campaigns in Uganda. Overall, this work highlights the critical need for novel approaches to sustain LLIN coverage between distribution campaigns, particularly towards the end of the 3-year cycle.

## Introduction

Uganda remains the country with the highest burden of malaria in the East African region, accounting for 5% of cases and 3% of deaths globally, with most deaths occurring among children under five years of age (under-fives) [1,2]. While there has been progress in the reduction of reported cases, malaria still accounts for approximately 20% of outpatient visits and inpatient admissions in Uganda [2,3].

Bed nets are highly effective and have been proven to have direct protection for the people using them, due to both a physical barrier and chemical lethality to mosquitoes [4]. As Uganda has the highest burden of malaria in Eastern Africa, maintenance of universal coverage, defined by the World Health Organization (WHO) as 1 long-lasting insecticidal nets (LLIN) for 2 people in the home, is a priority [5]. Uganda has been a leader in the distribution of LLINs since 2013 with mass distribution campaigns conducted every three years to establish universal coverage [6–8]. As a result, the proportion of households reportedly having at least one LLIN increased from 16% in 2006 to 80% in 2018, while universal coverage rates increased from 5% of households in 2006 to 54% in 2018 [9,10]. Yet towards the end of each three-year cycle, attrition due to damage and loss can leave households well below universal coverage targets with a resultant increase in malaria transmission intensity [11,12]. To maintain coverage between mass distribution campaigns, the World Health Organization (WHO) recommends continuous LLIN distributions through antenatal and immunization clinics [10].

In 2001, the Ugandan Ministry of Health (MoH) endorsed the creation of Village Health Teams (VHTs), composed of several community members selected by the community to serve as community health workers (CHWs). The core functions of these lay CHWs include health promotion and education. Select CHWs in some areas have also been trained according to the Ugandan Ministry of Health Integrated Community Case Management (iCCM) guidelines to treat under-fives in the community with common childhood illnesses such as malaria, pneumonia and diarrhea [13,14].

The main aim of this work was to address the LLIN coverage gap that emerges in the period after mass distribution campaigns and thereby reduce malaria transmission through implementation of a novel LLIN distribution strategy leveraging CHWs carrying out iCCM. As a first step towards this overall goal, this pilot study focused on assessing the feasibility and additional burden of integrating LLIN distribution into the current tasks of CHWs as well as understanding the level of need in the community. We hypothesized that iCCM-trained CHWs, who already provide first-line treatment for malaria in their communities, will be able to distribute LLINs to at-risk households that have fallen below target coverage rates without increased burden.

## Methods

### Study design

This was an open-label pilot study investigating the feasibility of supplemental, targeted LLIN distributions through a community health worker (CHW) led integrated community case management (iCCM) program in rural Uganda. To accomplish this, we implemented a distribution program in one village, while observing a second village to test our protocols and

assess the potential scale, reach, and feasibility of the proposed intervention, employing both quantitative and qualitative methodologies.

## Study site

This study was conducted in Maliba Sub-County of Kasese District of Western Uganda (Fig 1). Maliba is located in the foothills of the Rwenzori mountains with steep hillsides with elevations up to 2,500 meters. Western Uganda historically experiences two rainy seasons, from September to November and March to May; however, malaria transmission occurs year-round. Many communities at lower elevations sit along rivers. Subsistence agriculture and coffee growing is the main source of income.

According to the 2018–2019 Malaria indicator survey (MIS), which was conducted 1 year after a government-led mass distribution campaign, 58% of households reported meeting the universal coverage targets while 78% of household members could have slept under an LLIN if each LLIN in the household were used by up to two people [9]. Among household members,

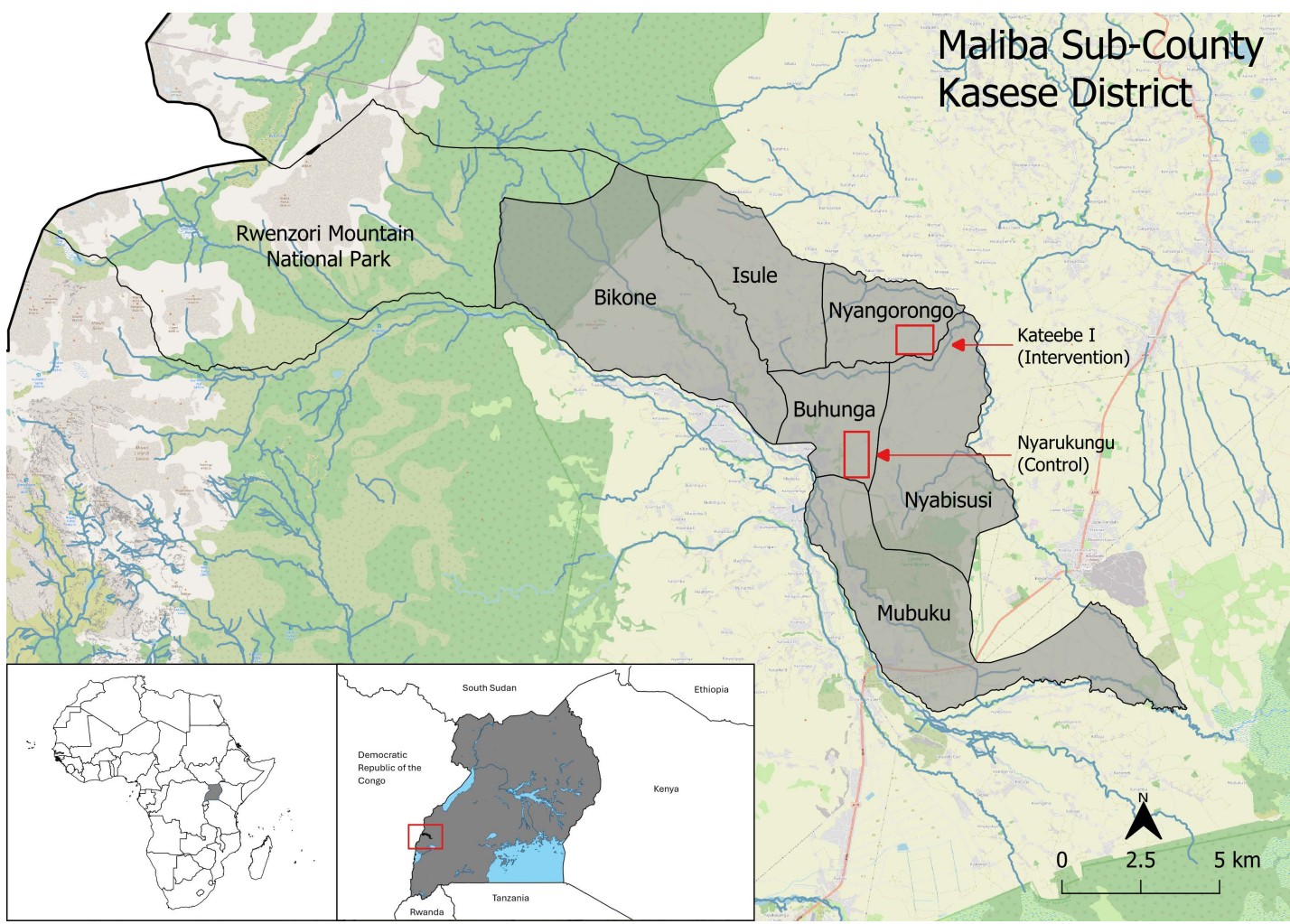

**Fig 1. Map of study area.** This map shows Maliba sub-district in light grey on the map panel. Each village enrolled in the study is indicated by the labeled areas outlined in red. Uganda is shaded dark grey in the map of Africa in the lower left corner. Maliba sub-district is colored black and marked by a red box on the map of Uganda in the bottom center of the map above.

68% of all individuals and 68% of children under the age of five slept under an LLIN the night before the survey. While the MIS reported a *P. falciparum* parasitemia rate (PfPR) of 7.3% for the larger Tooro district, we found PfPR rates in Bugoye Sub-County, which directly borders the study area, upwards of 30% among children 2 to 8 years old in some of the low-lying villages [15].

## Intervention

The two villages enrolled, Kateebe 1 (intervention) and Nyarukungu (control), were selected based on participation in the iCCM program supported by our research team, giving access to monthly data including the number of children assessed for malaria and diagnosed with malaria. Further, previous research conducted in the adjacent Bugoye sub-county indicated a PfPR among children 2 to 10 years of age over 30% for villages with similar terrain and geography as those selected for this study. Based on previous iCCM data, Kateebe 1 and Nyarukungu, had similar population sizes, number of iCCM visits, and proportion iCCM visits attributable to malaria during the previous year.

In brief, CHWs provide initial assessments, diagnosis via malaria rapid diagnostic test (mRDT), and treatment for children five years old and younger with uncomplicated malaria, pneumonia, and diarrhea. While CHWs are not medical professionals, they are considered "Level I" health providers and refer children with undifferentiated or complicated disease to higher-level health facilities. Children over the age of five are referred to a health facility and not treated by CHWs [14,16].

In addition to the existing workflow, we trained CHWs in the intervention village to supplement routine distribution programs (e.g., mass distribution campaigns, antenatal clinics) by distributing LLINs to households meeting the following criteria: (i) child presented to CHW who tested positive for malaria via a mRDT (Fig 2) and (ii) caregiver reported the number of LLINs within the household that fell below the WHO definition of universal coverage. CHWs calculated the number of LLINs to be distributed by dividing the number of individuals living in the home by two (universal coverage is one LLIN per two people) and subtracting the number of LLINs currently owned by the family. LLINs were only distributed to residents of the intervention village; however, CHWs in both villages assessed the number of LLINs needed in homes to reach full coverage.

## Quantitative data collection

Data were collected using two cross-sectional surveys before and after a three-month period of data collection by CHWs with the potential for households to appear in either the pre-intervention survey, post-intervention survey, or both surveys. Study staff working with the local CHWs attempted to visit all households in each village between June 8, 2023 and August 9, 2023. After consent was obtained, an adult caregiver completed a questionnaire assessing baseline demographic characteristics for everyone living in the home, characteristics of the dwellings, family structure, history of malaria, care seeking preferences, and LLIN ownership, quality, and use (S1 and S2 Appendices). Data were self-reported except for information regarding LLINs and characteristics of home construction, which were visually inspected by research staff.

Anthropometric measures were recorded for children aged 2 to 12 years old present at the home at the time of the survey, including temperature and Mid-Upper Arm Circumference (MUAC) for children 2 to 5 years old. The MUAC was used to determine if a child was malnourished with a cut off of 14.3 [17]. Children aged 2 to 10 were tested for malaria infection using HRP2-based mRDTs (Meril Meriscreen Malaria Pf/PAN Ag).

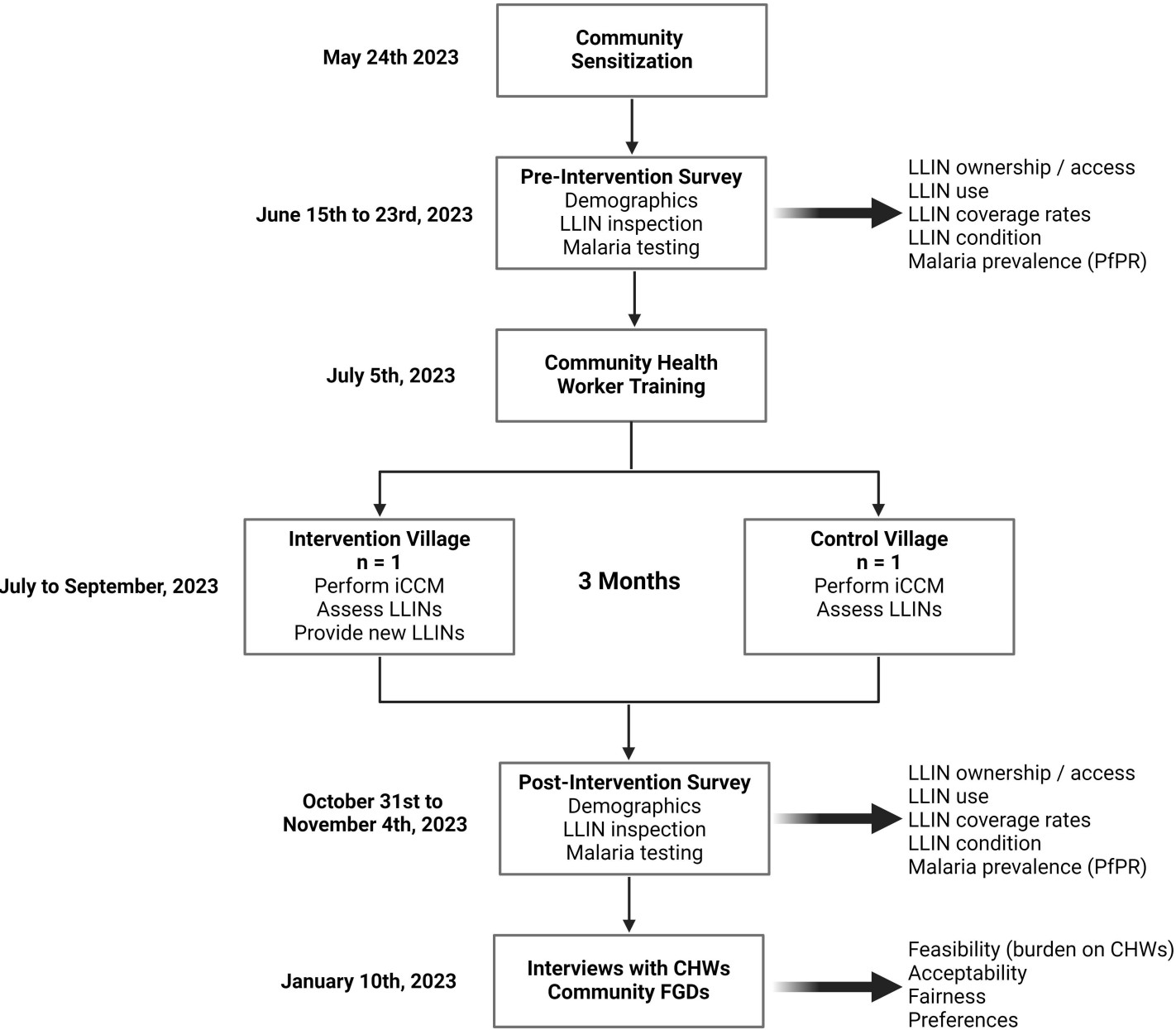

**Fig 2. Study protocol.** This flowchart depicts the steps in study rollout. Prior to the pre-intervention surveys, research teams visited the villages and local leaders to discuss the study. After these surveys, CHWs were trained in the new tasks, followed by 3 months of running the study. After conducting the post-intervention surveys, each CHW from both communities were interviewed to assess their experience with the intervention.

During the study period, July 2023 to September 2023, CHWs maintained tracking sheets of the numbers of children seen, tested, tested positive for malaria, and the number of LLINs distributed monthly. This tracking included only households enrolled in at least the pre-intervention survey.

End of study surveys were administered to the same households by study staff between October 31, 2023 to November 11, 2023. These surveys included the same questions as the baseline survey in addition to questions on malaria infection through the study period, new LLINs, and where the LLINs were obtained from. Temperature, MUAC, and incident malaria

infection during the study were collected. All data were collected using REDCap. Similar to the baseline study, data were predominantly self-reported outside of data regarding LLINs and household characteristics of home construction. The pre-intervention and post-intervention surveys are available in S1 Appendix and S2 Appendix, respectively.

For this study eight members of the nearby health facilities were trained over the course of two days to serve as research assistants and carry out data collection. The research assistants assisted in a day long training CHWs covering how to calculate the number of LLINs to distribute to each household and how to communicate with families about eligibility to receive LLINs. Minimal resources were needed to carry out this intervention, in addition to procurement of the LLINs, including stationery for data entry forms and calculations, meals and transportation for the training, and financial support for cell phone payments. While CHWs were provided a monthly allowance (20,000 Ugandan Shillings each) for their participation in the study, CHWs in the area work as volunteers.

## Qualitative data collection

After study follow-up concluded, an in-depth interview was conducted with each of the four participating CHWs, two from each village to better understand the community's acceptance to LLIN distribution and ease of calculations for LLIN distribution. In the intervention village additional questions about the burden and challenges associated with the intervention were asked as well as potential challenges with scaling up the intervention. Input about the importance of the interventions and suggestions were provided by the CHWs. Interviews lasted from 35 to 45 minutes for CHWs in the control village and 45 to 60 minutes in the intervention village.

## Analysis

We employed a widely used framework to assess the feasibility and acceptability of the intervention [18]. The primary outcome of interest was the proportion of iCCM clients eligible to receive LLIN, defined as RDT positive children living in a home below universal coverage. Secondary outcomes of interest include the total number of LLINs distributed by CHWs in the intervention village, the change in PfPR among children 2-10 years old between the pre- and post-intervention surveys, the status of homes across both arms with respect to universal coverage before and after the intervention, the proportion of people who slept the prior night under an LLIN and community acceptance of the intervention. All variables were measured as continuous variables other than status of households regrading universal coverage which was measured as a three-level categorical variable defined as owning no LLINs, owning LLINs but not enough to reach the universal coverage threshold, and owning enough LLINs to reach the universal coverage threshold.

This study was not powered for statistical tests comparing the intervention and control villages. All results are reported in descriptive tables providing counts and percentages for categorical data and as medians and IQR for continuous data [19,20]. In Table 1, Z-tests were used to test proportions between the control and intervention groups and Wilcoxon rank sum tests to compare medians using an alpha level of 0.05. All data management and analysis were conducted in R.

Qualitative data obtained from interviews with CHWs were summarized. No qualitative analysis techniques were used due to the small sample size (n = 4) and short length of interviews.

## Ethics

Ethical approval for the study was obtained from the institutional review boards of the University of North Carolina at Chapel Hill (22-2947) and the Mbarara University of Science and

**Table 1. Population demographics.**

| | Kateebe 1 Intervention (N (%)) | Nyarukungu Control (N (%)) | Total (N (%)) |
|---|---|---|---|
| | 153 Households | 180 Households | 333 Households |
| **Age** | | | |
| Adults (≥18 years) | 363 (42.7) | 417 (41.2) | 780 (41.9) |
| Children | | | |
| Age 13 to 18 years | **114 (13.4)** | **191 (18.9)** | 305 (16.4) |
| Age 5 to 12 years | **234 (27.5)** | **236 (23.3)** | 470 (25.2) |
| Age <5 years | 140 (16.5) | 167 (16.5) | 307 (16.5) |
| **Household Measures** | | | |
| Household size (median, IQR) | 5 (4–6) | 5 (4–7) | 5 (4–7) |
| Adults per household (median, IQR) | 2 (2–3) | 2 (2–3) | 2 (2–3) |
| Children per household (median, IQR) | 3 (2–4) | 3 (2–5) | 3 (2–4) |
| Earthen or sand floor | 141 (92.2) | 160 (88.9) | 301 (90.4) |
| Electricity in the house | **152 (99.3)** | **170 (94.4)** | 322 (96.7) |
| Access to piped water | **109 (71.2)** | **73 (40.6)** | 182 (54.7) |
| Own a mobile phone | **135 (88.2)** | **144 (80.0)** | 279 (83.8) |
| Own livestock | 103 (67.3) | 116 (64.4) | 219 (65.8) |
| Have a bank account | **133 (86.9)** | **135 (75.0)** | 268 (80.5) |
| **Adults** | | | |
| Age (median, IQR) | 34 (25–49) | 36 (25–50) | 35 (25–50) |
| Male | 153 (43.3) | 182 (44.6) | 335 (44.0) |
| Married[1] | 270 (91.5) | 308 (90.1) | 578 (90.7) |
| **Highest level of school [1]** | | | |
| No school | 33 (11.2) | 42 (12.2) | 75 (11.8) |
| Primary | 200 (67.8) | 244 (71.2) | 444 (69.6) |
| Secondary | 60 (20.3) | 54 (15.7) | 114 (17.9) |
| University | 2 (0.7) | 3 (0.9) | 5 (0.8) |
| **Children[2]** | | | |
| Age (median, IQR) | 6 (2–9) | 6 (3–9) | 6 (3–9) |
| Male[1] | 159 (45.6) | 157 (42.4) | 316 (43.9) |
| Malnourished (MUAC <14.3) [1,3] | 16 (17.6) | 10 (9.6) | 26 (13.3) |
| **Birth location[1]** | | | |
| Home | 85 (24.6) | 70 (19.1) | 155 (21.6) |
| Government health facility | **242 (69.3)** | **290 (79.0)** | 532 (74.3) |
| Government hospital | 15 (4.3) | 7 (1.9) | 22 (3.1) |
| Private clinic or hospital | **8 (2.3)** | **1 (0.3)** | 9 (1.3) |
| Hospitalized in the last 12 months from malaria (Children ≤ 18) | **79 (16.2)** | **128 (21.5)** | 207 (19.1) |

This table provides a description of the study population across the intervention (Kateebe 1) and control village (Nyarukungu) prior to start of the intervention. Total counts were included along with percentages in parenthesis unless otherwise specified as the median and interquartile range (IQR). Data on marital status and education level are only available for the head of the household and second adult, not all adults living in the home. Bolded values denote statistically significant values using an alpha of 0.05.

[1] Denotes a different sample size was used due to missing data than the total number of children or adults referenced at the top of the table. Missing data can be seen in S1 Table.

[2] Only includes children 12 years old and younger

[3] MUAC was only measured for children 2 to 5 years old.

Technology (2023-798). Written informed consent was obtained from heads of each household in the local language. If participants could not write, fingerprints were used in place of a signature.

## Results

### Baseline demographics

Between June 8 and August 9, 153 and 180 households were enrolled in the intervention and control villages, respectively, representing 851 and 1011 individuals. All households with an adult present agreed to participate in the study. Characteristics of the study populations are shown in Table 1. Baseline demographics were generally similar between the two villages. Households were approximately the same size with a median of 5 (IQR: 4-7) members across both communities with similar distributions of children and adults.

Households in Kateebe 1 reported higher rates of access or ownership to resources compared to Nyarukungu including electricity in house (99.3% vs 94.4%), access to piped water (71.2% vs 40.6%), owned a mobile phone (88.2% vs 80.0%), and had a bank account (86.9% vs 75.0%), respectively. Most of the adult population completed only primary school. About 91% of adults were married.

The median age of adults was 35 years and 6 years for children across both villages. Using the results of the MUAC measurement, a higher percentage of children under five years of age in the intervention village were malnourished compared to the control village (17.6% vs 9.6%). While most children in both villages were born in a government health facility (74.3%) about a fifth of the population (21.6%) was born at home. At baseline, more children had been hospitalized in the control village, however, 19.1% of all children in the study population had been hospitalized from malaria in the past year.

### Implementation

A total of 169 children under five years of age in the intervention village and 171 children under five years of age in the control village presented to their respective CHWs for evaluation from July to September 2023, with 47.9% of visits occurring in September (Table 2). Of the children evaluated by the CHWs, 102 of 169 (60.3%) and 112 of 171 (65.5%) were tested for malaria and of those tested, 62 (60.7%) and 71 (63.3%) were positive in the intervention and control villages, respectively. All 100 LLINs were distributed to 53 households of the 62 eligible to receive LLINs while the remaining 9 children eligible did not receive LLINs due to shortages (Table 2). Multiple LLINs were given to each household as needed to reach universal coverage. Therefore, the 100 LLINs from the study did not cover 100 households.

Most children who tested positive for malaria were also eligible to receive at least one LLIN from the CHW. Only three households were not eligible to receive LLINs because they were at universal coverage (Table 2). The median number of LLINs required to bring eligible households to universal coverage was 3 (IQR: 2 - 4). LLINs were distributed with increasing frequency from July to September with 48.0% of LLINs given out in September alone, at the start of the second rainy season. Care-cascades are also represented in S1 and S2 Figs for the intervention and control villages, respectively.

### Pre and post intervention statistics

The post-intervention survey was conducted in October and November 2023. During this survey, the study team reached a total of 109 and 149 households in the intervention and control villages, respectively. There was a reduction of 44 and 31 households enrolled from

**Table 2. Cascade for intervention data.**

| | Kateebe 1 Intervention (N (%)) | Nyarukungu Control (N (%)) | Total (N (%)) |
|---|---|---|---|
| | 140 Children[1] | 167 Children[1] | 307 Children[1] |
| Number of children seen by the CHW for any iCCM condition (Malaria, Pneumonia and Diarrhea) | | | |
| July | 43 (25.4) | 48 (28.1) | 91 (26.8) |
| August | 82 (48.5) | 81 (47.4) | 163 (47.9) |
| September | 44 (26.0) | 42 (24.6) | 86 (25.3) |
| Total | 169 | 171 | 340 |
| Number of children tested for malaria | | | |
| July | 30 (29.4) | 32 (28.6) | 62 (29.0) |
| August | 44 (43.1) | 51 (45.5) | 92 (43.0) |
| September | 31 (30.4) | 29 (25.9) | 60 (28.0) |
| Total | 102 | 112 | 214 |
| Number of children positive for malaria | | | |
| July | 21 (33.8) | 24 (33.8) | 45 (33.8) |
| August | 23 (37.1) | 31 (43.7) | 54 (40.6) |
| September | 18 (29.0) | 16 (22.5) | 34 (25.6) |
| Total | 62 | 71 | 133 |
| Number of children eligible to receive an LLIN [2] | 62 | 68 | 130 |
| Median number of LLINs missing/short per household (IQR) | 3 (2.5–4.5) | 2 (2–3) | 3 (2–4) |
| Number of children given LLINs | 53 | NA | NA |
| Number of LLINs distributed | | | |
| July | 14 (14.0) | NA | NA |
| August | 38 (38.0) | NA | NA |
| September | 48 (48.0) | NA | NA |
| Total | 100 | NA | NA |

This table provides counts of children moving through the cascade of care and intervention from being taken to the community health worker (CHW) to receiving an ITN. Only children five years old and younger were included because this is the age limit for children being seen by (CHWs). Total counts were included along with percentages in parenthesis. As Nyarukungu was the control village and did not receive the intervention, children were not eligible to receive ITNs, therefore the number of children receiving ITNs and the number of ITNs distributed were not applicable (NA) for analysis.

iCCM – integrated community case management

[1] The number of children enumerated by the study may be an undercount of children living in the communities. CHWs recorded more children in the village than were included in our baseline survey, which is why these numbers do not match the total number of children seen by a CHW for any iCCM condition. Children from other villages and repeat visits were excluded from CHW counts.

[2] Eligible children were those living in households that did not meet the universal coverage threshold of 2 people per ITN.

the post-intervention survey compared to the pre-intervention survey in the intervention and control villages, respectively. Not all households surveyed in the pre-intervention wave were reached in the post-intervention survey most often because household members were not present at the home when visited by research teams during data collection or additional follow-up attempts. There was a reported increase from 4.0% to 8.3% of households meeting universal coverage in the intervention village, compared to a decrease from 7.8% to 1.8% of households in the control village (Table 3). Similarly, the intervention village saw a marked increase in the number of children under the age of 5 who slept under an LLIN the previous night from 15.7% to 31.6% compared to the decrease seen in the control village from 29.1%

**Table 3. Pre-Post intervention data.**

| | Kateebe 1 Intervention Village (N (%)) | | Nyarukungu Control Village (N (%)) | |
|---|---|---|---|---|
| | Pre | Post | Pre | Post |
| **Sample sizes** | | | | |
| Households | 153 | 109 | 180 | 149 |
| Total individuals | 837 | 720 | 1007 | 951 |
| Adults | 363 (43.4) | 303 (42.1) | 417 (41.4) | 379 (39.9) |
| Children (total)[1] | 474 (56.6) | 417 (57.9) | 590 (58.6) | 572 (60.1) |
| Children aged 2 to 10 | 278 (33.2) | 234 (32.5) | 300 (29.8) | 300 (31.5) |
| Children aged 2 to 10 tested for malaria (PfPR) | 233 (83.8) | 153 (65.4) | 250 (83.3) | 260 (86.7) |
| Children under the age of 5 | 140 (16.7) | 122 (16.9) | 167 (16.6) | 158 (16.6) |
| **Among all households** | | | | |
| Level of coverage | | | | |
| No LLINs | 114 (75.5) | 111 (77.1) | 124 (69.3) | 87 (77.7) |
| Below universal coverage | 31 (20.5) | 21 (14.6) | 41 (22.9) | 23 (20.5) |
| Meets universal coverage | 6 (4.0) | 12 (8.3) | 14 (7.8) | 2 (1.8) |
| Households reporting incident malaria infection during study period (any households) [2] | NA | 58 (54.2) | NA | 126 (85.1) |
| Prevalent PfPR among children 2-10 years of age[3] | 21 (9.0) | 33 (21.6) | 90 (36.0) | 81 (31.2) |
| Change in PfPR among children 2-10 years of age[3] | 12 (12.6) | | -9 (-4.8) | |
| **Among household that have LLINs:** | | | | |
| Proportion of household members who slept the prior night under a LLIN (median (IQR)) | 0.6 (0.3–0.8) | 1 (0.8–1) | 0.9 (0.5–1) | 0.8 (0.4–1) |
| Reasons why household members did not sleep under an LLIN[4] | | | | |
| Not enough LLINs | 23 (92.0) | 15 (100) | 23 (95.8) | 11 (91.0) |
| LLIN has too many holes or is too damaged | 2 (8.0) | 0 (0) | 1 (4.2) | 1 (9.0) |
| Too hot, uncomfortable, or don't like the smell | 0 (0) | 0 (0) | 0 (0) | 0 (0) |
| No place to hang it | 0 (0) | 0 (0) | 0 (0) | 0 (0) |
| Not many mosquitoes/ low risk for malaria | 0 (0) | 0 (0) | 0 (0) | 0 (0) |
| Proportion of household LLINs with major damage (median (IQR))[5] | 0 (0–0) | 0 (0–0) | 0 (0–0) | 0 (0–0) |
| Proportion of LLINs were used last night (median (IQR)) | 1 (1–1) | 1 (1–1) | 1 (1–1) | 1 (1–1) |
| Under-fives who slept the prior night under a LLIN | 20 (15.7) | 36 (31.6) | 44 (29.1) | 15 (10.5) |
| Households receiving new LLINs from any source | NA | 29 (27.1) | NA | 6 (4.1) |
| Number of LLINs received from CHW by a household (median (IQR)) | NA | 3 (2 – 4) | NA | 0 (0) |
| Incident malaria infection during study period (since receipt of new LLINs) | NA | 3 (10.7) | NA | 0 (0) |

This table provides counts and percentages for individuals and households before and after the intervention in both the intervention (Kateebe 1) and control villages (Nyarukungu). Some data pertain only to the post intervention timepoint, and, therefore, were not applicable (NA) at the pre intervention timepoint. Total counts were included along with percentages in parenthesis. Percentages for adults, total children, and specific age groups of children are out of all participants. Percentages of children tested for malaria are out of children aged 2 to 10 years old.

[1] 14 children from the intervention village and 4 children from the control village were excluded due to missing data for malaria related variables

[2] Denotes a different sample size was used due to missing data than the total number of children or adults referenced at the top of the table. Missing data can be seen in S2 Table.

[3] PfPR - Plasmodium falciparum parasite rate

[4] More than one answer could be selected by households. Only includes households that had LLINs and household members that did not sleep under an LLIN

[5] Major damage is defined as no tears or holes >2 cm (i.e., larger than thumb)

to 10.5%. Not owning enough LLINs was almost exclusively reported as the reason that not everyone slept under an LLIN the previous night.

The number of households reporting incident cases of malaria at the end of the study period was over 50% higher in the control village at 85.1% of households compared to only 54.2% of

households in intervention village. However, the intervention village had an increase in PfPR among children while the control village reported a decrease compared to baseline testing.

In the intervention village, 27.1% of homes received new LLINs, 93.3% of which came from CHWs through the study intervention. Only 10.7% of homes that received new LLINs reported a malaria infection after receiving the new LLINs.

## CHWs responses to the intervention

**Burden of work.** CHW in the intervention village indicated that the additional tasks for the intervention took between 20 to 30 minutes in addition to their usual tasks. None of the CHWs reported the additional tasks as impacting their current work and indicated that it was a manageable workload.

**Challenges.** The only challenge reported was that residents of the control and neighboring villages wanted LLINs but the CHWs were unable to distribute LLINs to them under the study protocol. All four CHWs involved in the project recommended that the intervention be implemented and expanded to other villages. While residents of both villages had a high acceptance of the intervention, many residents testing negative for malaria were still interested in receiving LLINs. The most common challenge to program expansion suggested by the CHWs was the need for community engagement. Effort needs to be put into talking with community members and local leaders to ensure acceptance of the program.

## Discussion

Our pilot study demonstrates the feasibility and acceptability of targeted LLIN distributions leveraging the existing ICCM structure to supplement national distribution campaigns in Uganda. Of the children who tested positive for malaria, 100% and 96% of the children in the intervention and control villages were eligible to receive at least on LLIN, showing the need to additional LLIN distribution. Further, the complete distribution of LLINs given to the CHWs for this study indicates the feasibility of this method but may require additional effort to ensure the LLIN with CHWs remains sufficient.

Although the baseline level of LLIN coverage was substantially below our initial estimates, which resulted in CHWs having to distribute more nets than anticipated, the CHWs did not perceive the additional workload as overly burdensome and favored more widespread implementation. Importantly, we observed that the distributed LLINs were present in the home during the post-intervention survey and were reflected in the increased proportion of children under five years of age sleeping under a net in the intervention as compared to the control village. We saw a one-fold increase in LLIN usage in children under 5 in the intervention village and a two-fold decrease in the control village, with similar directional trends in medians for the overall number of household members sleeping under an LLIN. This increase is likely driven by increased access and quality of LLINs in the home through the intervention while LLINs continued to deteriorate in the control village. Overall, this work highlights the critical need for novel approaches to sustain LLIN coverage between distribution campaigns, particularly towards the end of the 3-year cycle. It is notable that 12 households were found to reach the universal coverage threshold during the post-intervention survey in the intervention village although 29 households reported receiving LLINs from the study. We hypothesize that over the study period, there was continued attrition of LLINs from the previous national distribution four years earlier and that the loss of those LLINs could have led to households falling back below the threshold. Secondly, it is possible that many of the households receiving LLINs from the study were only captured in the pre-intervention survey, estimated to be 24 households, leading to over representation of those below the threshold.

The approach explored in this study has several advantages. First, we leveraged the existing network of CHWs. Not only are these individuals experienced with providing algorithm-based care of young children under the iCCM program, through the use of clear flowcharts for diagnosis and treatment, but they are also trusted members of local communities [21]. Second, our approach entails continuous LLIN distribution, thereby addressing gaps that can occur between national distribution campaigns due to logistical disruptions or external events such as the COVID-19 pandemic. Lastly, by restricting eligibility to children who test positive for malaria we are targeting the intervention to households that are likely at the highest risk of malaria. This strategy may ultimately prove more resource sparing and ultimately cost-effective. For example, in our prior work, we found low malaria transmission (PfPR =< 2%) in many of the highest elevation (>1600 m) villages [22]. Efforts to maintain universal coverage in these areas may be relatively low yield, although this merits further study.

Potential challenges with scaling up the intervention could include maintaining an appropriate stock of LLINs for distribution. This may not only involve issues of supply chains, but also storage capacity as CHWs would be required to store the LLINs in their homes, most of which are relatively modest in size. Reassuringly, the participating CHWs did not report this concern during interviews, although the limited scale and duration of our pilot may not have stressed the CHWs in a way that full implementation may do. There may also be challenges maintaining the supply of LLINs if LLIN attrition rates are high. As this was a pilot feasibility study with a limited, 3-month duration, we did not assess attrition of the existing or newly provided LLINs [23–25].

From an acceptability standpoint, there was broad support and perceived importance of taking additional steps to reduce the malaria burden for children. Similar support for additional use and distribution of LLINs by CHWs as community level interventions were seen in other communities in Uganda [26–28]. These studies also reported LLINs as preferred by community members compared to other methods of prevention because LLINs are free and easy to clean. However, some community members preferred screening windows, as it provided the longest-lasting barrier [29,30]. While LLINs do require replacement, screening windows on every home may be more challenging due to a lack of an existing framework for implementation at scale, difficulty and cost as compared to the ease of LLIN distribution [24,31,32]. Nevertheless, it can complement LLIN use.

Our study has a number of strengths including high levels of participation, a near census of the participating communities, and longitudinal data on community interface with CHWs. However, there are also important limitations. First, as a pilot feasibility design, our study was not intended or adequately powered to test outcomes related to the effectiveness of the intervention compared to the control [20,33]. While it is encouraging to see increases in the number and proportion of households reporting universal coverage targets and children under five years of age sleeping under a LLIN in the intervention village, given the small sample and potential for confounding, this should not be interpreted as evidence of effect. Similarly, we note that $PfPR_{2-12}$ increased from pre- to post-intervention in the intervention village, whereas it decreased in the control village. We surmise that this is attributable to factors external to the intervention, including changes in the local ecology of the vectors with the transition from dry to wet seasons. These dynamics can be better understood by observing both seasons and their transitions. A longer follow-up period could also shed more light on the impact of this distribution method as more households accumulate at universal coverage. To assess the effectiveness of the intervention to sustain coverage and prevent malaria, a larger and more rigorous design such as a cluster randomized controlled trial will be required. Due to the limitations of the pilot design, we did not have the resources to carry out larger focus groups with

community members and instead relied on reports made to the CHWs. Therefore, we were not able to fully assess community sentiments about the program directly.

Secondly, these two communities were selected as a convenience sample and not balanced by potential covariates of interest. Therefore, noticeable differences were seen with the intervention village having a higher overall SES and a lower incidence rate of malaria and malaria related hospitalizations than the control village, which are likely related given the established connection between SES and malaria [34]. Further, the control village was located at a lower elevation and in closer proximity to a stream. While we do not believe these factors had an impact on assessing feasibility, it is possible that more LLINs would have been distributed in the control village had it been designated the intervention village, conveying a greater need for the novel intervention but possibly overstating the impacts of the program on reductions in malaria cases.

Thirdly, the study took place at the very end of a 3-year LLIN mass distribution cycle, which had been further delayed by the pandemic. Therefore, LLIN ownership rates were extremely low, which likely exaggerated the potential impact of the intervention, specifically in regard to the number of households eligible to receive LLINs through the program.

Lastly, while we were able to identify a number of the distributed nets during the post-intervention survey, we cannot be sure that all were used as intended. More robust measures of long-term use should be incorporated into future studies.

## Conclusion

CHWs play a pivotal role in reaching rural and hard to access communities, with previous success in helping to reduce the burden or malaria through early case management. As hypothesized, our study showed that CHWs can effectively distribute supplemental LLINs in remote communities without increased burden and high acceptability from community members. As this is a proof of concept for novel way to sustain coverage, future trials needed to show (i) it does sustain coverage and (ii) it actually reduces malaria transmission.

## Supporting information

**S1 Fig.  LLIN care cascade from control village.** Care cascade indicating the number of children who proceed to each step of the screening process to determine which children are eligible to receive a LLIN through the study in the control village, Nyarukungu. Children meeting each criterion are included in the sample size in each box. Children not meeting the criteria are removed with bolded arrows including the number of children removed and the reason for removal before moving onto the next criterion. The sample size (n) provided in the blue box represents the number of children enumerated from the control village. The starting number of children (n) in the first white box represents the number of visits to a CHW. The number of children enumerated by the study may be an undercount of children living in the communities and was lower than the number of children seen by the CHWs. (TIF)

**S2 Fig.  LLIN care cascade from intervention village.** Care cascade indicating the number of children who proceed to each step of the screening process to determine which children would receive a LLIN through the study in the intervention village, Kateebe 1. Children meeting each criterion are included in the sample size in each box. Children not meeting the criteria are removed with bolded arrows including the number of children removed and the reason for removal before moving onto the next criterion. The sample size (n) provided in the blue box represents the number of children enumerated from the intervention village. The starting number of children (n) in the first white box represents the number of visits to a CHW. The

number of children enumerated by the study may be an undercount of children living in the communities and was lower than the number of children seen by the CHWs.
(TIF)

**S1 Table. Missing data for population demographics.** This table provides counts and percentages for missing values out of all eligible participants per cell for the description of the study population as presented in Table 1. Percentages for missing data on marriage status and education level are calculated from all adults in the study, not just the head of the household and second adult as in Table 1.
(DOCX)

**S2 Table. Missing data for pre-post intervention data.** This table provides counts and percentages for missing values out of all eligible participants per cell for individuals and households before and after the intervention in both the intervention (Kateebe 1) and control villages (Nyarukungu). Some data pertain only to the post intervention timepoint, and, therefore, were not applicable (NA) at the pre intervention timepoint. Total counts were included along with percentages in parenthesis. Percentages for adults, total children, and specific age groups of children are out of all participants. Percentages of children tested for malaria are out of children aged 2 to 10 years old.
(DOCX)

**S1 Appendix. Pre-intervention survey.**
(PDF)

**S2 Appendix. Post-intervention survey.**
(PDF)

**S1 Checklist. Inclusivity in global research.**
(DOCX)

## Acknowledgments

We wish to thank the residents of Kateebe 1 and Nyarukungu Villages who participated in the study. In addition, we recognize the support of the Kasese District Health Office who facilitated study implementation.

Previous publication. Preliminary results were presented at the Future of Malaria Research Symposium, held October 13, 2023, in Baltimore, Maryland and The Carolina Malaria Research Annual Symposium (CaMRAS) on October 3, 2023, in Research Triangle Park, North Carolina. The authors confirm that the submitted work has not otherwise been previously presented or published in any format

## Author contributions

**Conceptualization:** Rapheal Mbusa, Emmanuel Baguma, Emmanuel Ayebare, Raquel Reyes, Moses Ntaro, Edgar M. Mulogo, Ross M. Boyce.

**Data curation:** Annika K. Gunderson.

**Formal analysis:** Annika K. Gunderson, Rapheal Mbusa, Ross M. Boyce.

**Funding acquisition:** Ross M. Boyce.

**Investigation:** Rapheal Mbusa, Emmanuel Baguma, Emmanuel Ayebare, Dana Giandomenico, Moses Ntaro, Edgar M. Mulogo, Ross M. Boyce.

**Supervision:** Ross M. Boyce.

**Visualization:** Annika K. Gunderson.

**Writing – original draft:** Annika K. Gunderson, Rapheal Mbusa, Ross M. Boyce.

**Writing – review & editing:** Annika K. Gunderson, Rapheal Mbusa, Emmanuel Baguma, Emmanuel Ayebare, Dana Giandomenico, Raquel Reyes, Moses Ntaro, Edgar M. Mulogo, Ross M. Boyce.

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
