## [Decision Letter · Decision Letter 0]

16 Jul 2024

PGPH-D-24-01366

Targeted distribution of long-lasting insecticidal nets by community health workers to sustain household coverage: a pilot feasibility trial in western Uganda

Dear Dr. Boyce,

Thank you for submitting your manuscript to PLOS Global Public Health. After careful consideration, we feel that it has merit but does not fully meet PLOS Global Public Health’s publication criteria as it currently stands. Therefore, we invite you to submit a revised version of the manuscript that addresses the points raised during the review process.

We look forward to receiving your revised manuscript.

Kind regards,

Abhinav Sinha, M.D.

Academic Editor

Journal Requirements:

1. Please include a complete copy of PLOS’ questionnaire on inclusivity in global research in your revised manuscript. Our policy for research in this area aims to improve transparency in the reporting of research performed outside of researchers’ own country or community. The policy applies to researchers who have travelled to a different country to conduct research, research with Indigenous populations or their lands, and research on cultural artefacts. The questionnaire can also be requested at the journal’s discretion for any other submissions, even if these conditions are not met.  Please find more information on the policy and a link to download a blank copy of the questionnaire here: https://journals.plos.org/globalpublichealth/s/best-practices-in-research-reporting . Please upload a completed version of your questionnaire as Supporting Information when you resubmit your manuscript.

2. Please provide separate figure files in .tif or .eps format.

Additional Editor Comments (if provided):

Reviewers' comments:

Reviewer's Responses to Questions

**Comments to the Author**

1. Does this manuscript meet PLOS Global Public Health’s publication criteria ? Is the manuscript technically sound, and do the data support the conclusions? The manuscript must describe methodologically and ethically rigorous research with conclusions that are appropriately drawn based on the data presented.

Reviewer #1: Yes

Reviewer #2: Yes

2. Has the statistical analysis been performed appropriately and rigorously?

Reviewer #1: No

Reviewer #2: Yes

3. Have the authors made all data underlying the findings in their manuscript fully available (please refer to the Data Availability Statement at the start of the manuscript PDF file)?

Reviewer #1: No

Reviewer #2: Yes

4. Is the manuscript presented in an intelligible fashion and written in standard English?

Reviewer #1: Yes

Reviewer #2: Yes

5. Review Comments to the Author

Reviewer #1: I have reviewed the manuscript titled "Targeted distribution of long-lasting insecticidal nets by community health workers to sustain household coverage: a pilot feasibility trial in western Uganda" and appreciate the thorough investigation into the implementation and impact of community health worker-led interventions. The authors' exploration of the feasibility and sustainability of targeted long-lasting insecticidal net distribution offers significant insights for public health practitioners and policymakers aiming to enhance malaria prevention strategies. However, several important issues require attention.

Title

The study does not exhibit the characteristics of a trial. It is recommended to use "pilot feasibility study" instead of "pilot feasibility trial" in the title.

Introduction

1. The malaria data presented in the introduction is outdated. Please update the data with the most recent malaria reports.

2. In the last line of the introduction paragraph, it is unclear whether the statement refers to Uganda specifically or to the global population in general. Please clarify and verify reference number 4.

3. In paragraph 2, line 77, a more recent citation should be provided to highlight the effectiveness of long-lasting insecticidal nets (LLIN) against malaria.

4. A reference for the WHO universal coverage statement is missing. According to WHO, "Universal coverage for malaria vector control is defined as universal access to and use of appropriate interventions by populations at risk of malaria." The term "universal coverage" is not suitable here and should be replaced with a more appropriate term. The current coverage rate, including accessibility and utilization attrition rate, and loss rate in Uganda should also be presented.

Study Aim and Methodology

1. The aim of the study provided does not align with the methodology and results sections. The LLIN coverage gap in Uganda should be explained in detail, referencing published literature. Additionally, the reduction of malaria transmission is not mentioned in the results or discussion sections and should be revised accordingly.

2. In the methodology, it is unclear how both the intervention and control villages were selected. The criteria for their selection need to be clearly explained.

3. The status of Nyagorongo and Bhugunga, from where the intervention and control villages were selected, should be explained. The reasons for choosing these areas, considering their socio-economic, ecological, and ethnic differences, as shown in Table 1, should be provided.

4. It is also unclear why a control village was included when the objective was to assess the feasibility of interventions by community health workers (CHWs). This should be justified.

5. The gap identified in the Malaria Indicator Survey (MIS) should be part of the introduction to justify the research question. Additionally, the PfPr rates for Maliba sub-county should be provided to support the study site selection.

Intervention Section

1. The criteria for site selection should be clearly written.

2. Figure 2 lacks a clear timeline, including community sensitization, pre-intervention survey, CHW training, iCCM performance, LLIN assessment and distribution, post-intervention survey, and interviews with CHWs (FGDs). These details, including the timeline (dates), human resources involved, and monitoring details, are crucial for understanding the feasibility of the interventions.

3. The questionnaire used for the pre-intervention and post-intervention surveys should be provided.

4. Despite the feasibility study design, the sample size for recruiting households (HHs) in both intervention and control arms should be calculated.

5. In qualitative data collection, the number of CHWs involved in FGDs, the number of FGDs conducted in intervention and control villages, and whether they were IDIs or FGDs should be clearly detailed.

Outcome Indicators

1. For the primary outcome indicator (proportion of iCCM clients (HH) eligible to receive LLIN), specify if there were any cutoffs. Similarly, for secondary outcome indicators, provide the cutoffs or criteria to check feasibility and acceptability.

Results Section

1. It is unclear why the authors considered the age of children as 5-18 years in the results section while testing for iCCM included 2-18 years. The pre-intervention survey considered data for below 18 years, whereas the outcome indicator was for families with 2-10-year-old malaria-positive cases. Clarify if there were significant differences in malaria cases between case and control villages and why they were considered for inclusion.

2. It is unclear how 100 LLINs covered 53 malaria-positive children HHs with missing and short LLINs per HH. Similarly, explain how all eligible HHs achieved universal coverage when 9 eligible children HHs could not receive LLINs.

3. In pre and post-intervention statistics, explain why fewer HHs were followed for the post-intervention survey.

4. When almost all HHs were covered with LLIN distribution, clarify why only 6.5% of HHs met universal coverage.

5. Table 3 is unclear, particularly why after distributing LLINs in 62 malaria-positive children's households (Table 2), there were no LLINs in 11 households in the post-intervention survey, and only 12 HHs had universal coverage.

6. In Table 3, the data in columns 13 and 14 is confusing. When the PfPr among children 2-10 years of age is 9%, clarify how the household reporting incident malaria infection at the pre-intervention is zero. Revise the data presentation of malaria rates in households pre and post-interventions for better clarity.

7. Present the qualitative data thematically.

Discussion

1. Revisit and discuss the feasibility and acceptability claims with primary outcome indicator data.

2. Explain what "algorithm-based care" entails.

3. The acceptability of interventions in the community is not properly assessed. This should be discussed in light of the results.

By addressing these comments, the manuscript will be improved in terms of clarity, comprehensiveness, and scientific rigor.

Reviewer #2: The net result is that in the intervention village there was an two-time increase in the usage of nets, whereas in the control village the same was reverse with huge decrease in the usage. Please explain the reasons. Also there is no information on the attrition rate. Please revise the MS accordingly

6. PLOS authors have the option to publish the peer review history of their article (what does this mean? ). If published, this will include your full peer review and any attached files.

**Do you want your identity to be public for this peer review?** For information about this choice, including consent withdrawal, please see our Privacy Policy .

Reviewer #1: No

Reviewer #2: **Yes: ** Dr Susanta Kumar Ghosh

While revising your submission, please upload your figure files to the Preflight Analysis and Conversion Engine (PACE) digital diagnostic tool, https://pacev2.apexcovantage.com/ . PACE helps ensure that figures meet PLOS requirements. To use PACE, you must first register as a user. Registration is free. Then, login and navigate to the UPLOAD tab, where you will find detailed instructions on how to use the tool. If you encounter any issues or have any questions when using PACE, please email PLOS at figures@plos.org. Please note that Supporting Information files do not need this step.

---

## [Decision Letter · Decision Letter 1]

6 Dec 2024

PGPH-D-24-01366R1

Targeted distribution of long-lasting insecticidal nets by community health workers to sustain household coverage: a pilot feasibility study in western Uganda

Dear Dr. Boyce,

Thank you for submitting your manuscript to PLOS Global Public Health. After careful consideration, we feel that it has merit but does not fully meet PLOS Global Public Health’s publication criteria as it currently stands. Therefore, we invite you to submit a revised version of the manuscript that addresses the points raised during the review process.

We look forward to receiving your revised manuscript.

Kind regards,

Abhinav Sinha, M.D.

Academic Editor

Journal Requirements:

Additional Editor Comments (if provided):

Reviewers' comments:

Reviewer's Responses to Questions

**Comments to the Author**

1. If the authors have adequately addressed your comments raised in a previous round of review and you feel that this manuscript is now acceptable for publication, you may indicate that here to bypass the “Comments to the Author” section, enter your conflict of interest statement in the “Confidential to Editor” section, and submit your "Accept" recommendation.

Reviewer #1: All comments have been addressed

Reviewer #2: All comments have been addressed

2. Does this manuscript meet PLOS Global Public Health’s publication criteria ? Is the manuscript technically sound, and do the data support the conclusions? The manuscript must describe methodologically and ethically rigorous research with conclusions that are appropriately drawn based on the data presented.

Reviewer #1: Yes

Reviewer #2: Yes

3. Has the statistical analysis been performed appropriately and rigorously?

Reviewer #1: Yes

Reviewer #2: Yes

4. Have the authors made all data underlying the findings in their manuscript fully available (please refer to the Data Availability Statement at the start of the manuscript PDF file)?

Reviewer #1: Yes

Reviewer #2: Yes

5. Is the manuscript presented in an intelligible fashion and written in standard English?

Reviewer #1: Yes

Reviewer #2: Yes

6. Review Comments to the Author

Reviewer #1: I appreciate the revisions made to the manuscript, and I found that most of my previous comments have been addressed satisfactorily. However, I would like to raise a few additional points for clarification:

1. I observed significant variability in the sample sizes presented across Tables 1, 2, and 3. This inconsistency makes it challenging to understand the reasons behind the differences. For instance, in Table 1, the total number of children is 488 in 154 households, whereas in Table 3, the number of children is 474 in 153 households. It is unclear why these figures do not match. Could you kindly clarify whether these households are the same or if they differ between Table 1 and Table 3? Additionally, the sample sizes for pre-intervention households in both intervention and control villages (table 3 )do not align with the data provided in Table 1. A detailed explanation for these discrepancies would be helpful. Furthermore, please ensure that the table names are placed above the tables for consistency.

2. I noticed that there is a discrepancy between the numbers mentioned in the text and those presented in Table 2. This should be addressed in the manuscript. It would be beneficial if the text explicitly references the corresponding table numbers to avoid confusion.

3. In Table 3, the sample size for the intervention village's post-intervention households is listed as 109, but in the LLIN coverage data, the figure provided is for 144 household. None of these totals align with the household numbers provided for either the intervention or control groups.

4. The details provided in lines 199-200 regarding funding are not necessary for the revised manuscript. I suggest omitting this information.

5. The statement “They assisted in a day-long training of CHWs covering how to calculate the number of LLINs to distribute to each household and how to communicate with families about eligibility to receive LLINs” is unclear. It is not specified who "they" refers to.

6. The significant difference in sociodemographic variables, particularly the access to piped water between the control and intervention villages, could potentially contribute to the observed high malaria incidence rate. This factor can be discussed in the discussion section to provide a comprehensive analysis.

Reviewer #2: Still my question on LLIN attrition is not addressed properly. Because attrition rate will decide the community acceptance and ultimately the overall impact on the disease

7. PLOS authors have the option to publish the peer review history of their article (what does this mean? ). If published, this will include your full peer review and any attached files.

**Do you want your identity to be public for this peer review?** For information about this choice, including consent withdrawal, please see our Privacy Policy .

Reviewer #1: No

Reviewer #2: **Yes: ** SUSANTA KUMAR GHOSH

While revising your submission, please upload your figure files to the Preflight Analysis and Conversion Engine (PACE) digital diagnostic tool, https://pacev2.apexcovantage.com/ . PACE helps ensure that figures meet PLOS requirements. To use PACE, you must first register as a user. Registration is free. Then, login and navigate to the UPLOAD tab, where you will find detailed instructions on how to use the tool. If you encounter any issues or have any questions when using PACE, please email PLOS at figures@plos.org. Please note that Supporting Information files do not need this step.

---

## [Editor Report · Decision Letter 2]

12 Dec 2024

Targeted distribution of long-lasting insecticidal nets by community health workers to sustain household coverage: a pilot feasibility study in western Uganda

PGPH-D-24-01366R2

Dear Dr Boyce,

We are pleased to inform you that your manuscript 'Targeted distribution of long-lasting insecticidal nets by community health workers to sustain household coverage: a pilot feasibility study in western Uganda' has been provisionally accepted for publication in PLOS Global Public Health.

Best regards,

Abhinav Sinha, M.D.

Academic Editor